Locating ligand binding sites in G-protein coupled receptors using combined information from docking and sequence conservation

Vidad Ashley Ryan 1
http://orcid.org/0000-0003-1644-5525 Macaspac Stephen 1
http://orcid.org/0000-0002-6415-1938 Ng Ho Leung 2 hng@ksu.edu
1 Department of Chemistry, University of Hawaii at Manoa , Honolulu, Hawaii , United States of America
2 Department of Biochemistry and Molecular Biophysics, Kansas State University , Manhattan, Kansas , United States of America
Silva Pedro
Electronic publication date: 2021 Sep 24
Publication date: 2021
Volume: 9
Electronic Location ID: e12219
Received 2020 Sep 16; Accepted 2021 Sep 6
Copyright: © 2021 Vidad et al.
Copyright year: 2021
Copyright holder: Vidad et al.
License: This is an open access article distributed under the terms of the Creative Commons Attribution License, which permits unrestricted use, distribution, reproduction and adaptation in any medium and for any purpose provided that it is properly attributed. For attribution, the original author(s), title, publication source (PeerJ) and either DOI or URL of the article must be cited.
License URL: https://creativecommons.org/licenses/by/4.0/

Keywords: Ligand binding sites, GPCR, GPER, Drug binding sites, Bioinformatics, Modeling, Docking, Structure conservation, Molecular evolution

Funding: Geist Foundation 1833181 University of Hawaii at Manoa This work was funded by the Victoria S. and Bradley L. Geist Foundation (Ho Leung Ng), NSF CAREER Award 1833181 (Ho Leung Ng), and the Undergraduate Research Opportunities Program at the University of Hawaii at Manoa (Ashley R Vidad, Stephen Macaspac). The funders had no role in study design, data collection and analysis, decision to publish, or preparation of the manuscript.

==============================
GPCRs (G-protein coupled receptors) are the largest family of drug targets and share a conserved structure. Binding sites are unknown for many important GPCR ligands due to the difficulties of GPCR recombinant expression, biochemistry, and crystallography. We describe our approach, ConDockSite, for predicting ligand binding sites in class A GPCRs using combined information from surface conservation and docking, starting from crystal structures or homology models. We demonstrate the effectiveness of ConDockSite on crystallized class A GPCRs such as the beta2 adrenergic and A2A adenosine receptors. We also demonstrate that ConDockSite successfully predicts ligand binding sites from high-quality homology models. Finally, we apply ConDockSite to predict the ligand binding sites on a structurally uncharacterized GPCR, GPER, the G-protein coupled estrogen receptor. Most of the sites predicted by ConDockSite match those found in other independent modeling studies. ConDockSite predicts that four ligands bind to a common location on GPER at a site deep in the receptor cleft. Incorporating sequence conservation information in ConDockSite overcomes errors introduced from physics-based scoring functions and homology modeling.

Introduction

GPCRs (G-protein coupled receptors) are the largest family of drug targets and the targets of >30% of all drugs. Because they are membrane proteins with flexible and dynamic structures, biochemical and crystallography experiments are difficult. Only ~87 GPCRs out of ~800 in the human genome have been crystallized despite their great pharmacological importance. GPCR homology modeling remains challenging due to conformational flexibility and the abundance of flexible loops (Lai et al., 2017). Crystal structures have shown that the large majority of ligands bind in the large central, extracellular cavity of GPCRs, but the specific binding sites in the cavity can vary widely between different ligands even for the same or closely related receptors (Wacker, Stevens & Roth, 2017; Chan et al., 2019).

Various computational approaches have been used to predict ligand binding sites in G-protein coupled receptors. Traditional docking methods compute the lowest energy pose of a ligand fit to a receptor surface. Such methods are highly dependent on the form of the energy scoring function and accuracy of the receptor model structure (Katritch et al., 2010; Katritch & Abagyan, 2011; Shoichet & Kobilka, 2012; Weiss et al., 2016; Lim et al., 2018). These methods have been used to identify ligand binding sites and build pharmacophores for GPCRs (Kratochwil et al., 2011; Sanders et al., 2011; Tang et al., 2012), but the lack of diverse GPCR crystal structures presents serious challenges to using docking methods for identification of ligand binding sites. In particular, few crystal structures of non-class A GPCRs have been determined. Moreover, homology models usually cannot be used to identify ligand binding sites or for docking without extensive optimization, such as with advanced molecular dynamics sampling methods (Katritch et al., 2010; Lai et al., 2017; Zou, Ewalt & Ng, 2019). An underappreciated feature that can be used to predict ligand binding sites is surface or sequence conservation. Binding sites for particular ligands are often conserved, and systematic sequence variation can encode ligand specificity (Capra & Singh, 2007; Kalinina, Gelfand & Russell, 2009; Wass & Sternberg, 2009). While highly conserved receptors often share similar ligand binding sites, such direct relationships often do not apply between less conserved receptors. Yet, the massive abundance of genomic data for GPCRs can provide strong constraints for possible ligand binding sites even without chemical or structural information (Madabushi et al., 2004; Sanders, 2011; Levit et al., 2012). The binding sites for synthetic, non-physiological ligands can also be identified as they often share some or even most of their binding sites with physiological ligands (Wacker, Stevens & Roth, 2017). However, binding site conservation information alone usually cannot predict ligand binding sites, as often, very large parts of the receptor are highly conserved, including key structural elements.

There has been less research on methods that combine information from chemical interactions, geometric surface analysis, and bioinformatics. Hybrid strategies, such as Concavity (Capra et al., 2009), have demonstrated superior performance in predicting ligand binding sites compared to single-mode approaches. Concavity scores binding sites by evolutionary sequence conservation, as quantified by the Jensen–Shannon divergence (Capra & Singh, 2007), and employs geometric criteria of size and shape. Here, we describe a new hybrid strategy we have developed, called ConDockSite, to predict ligand binding sites from combined information from surface conservation and docking calculations. We compare our results with those previously published using purely docking-based and other hybrid methods (Arnatt & Zhang, 2013; Méndez-Luna et al., 2015). ConDockSite is not intended to be used for docking, i.e., predicting ligand binding poses, which are highly sensitive to small structural details in crystal structures. We demonstrate the effectiveness of ConDockSite for identifying ligand binding sites for the two best characterized class A GPCRs with known crystal structures, the β2 adrenergic and A2A adenosine receptors. We then demonstrate the effectiveness of ConDockSite with high quality GPCR homology models.

Finally, we apply ConDockSite to predict the hypothetical binding sites of four ligands to the less characterized class A G-protein coupled estrogen receptor (GPER, formerly known as GPR30), a membrane-bound estrogen receptor. GPER is proposed to mediate rapid estrogen-associated effects, cAMP regeneration, and nerve growth factor expression (Kvingedal & Smeland, 1997; Carmeci et al., 1997; O’Dowd et al., 1998; Filardo et al., 2002; Kanda & Watanabe, 2003). GPER is known to bind estradiol and the estrogen receptor inhibitors, tamoxifen and fulvestrant, that are used to treat breast cancer (Fig. S1). Recently, GPER-specific ligands G1 and G15 were discovered (Bologa et al., 2006; Dennis et al., 2009). G1 and G15 are structurally similar, differing by only an acetyl group. G1 is an agonist, whereas G15 is an antagonist. No crystal structure of GPER is available, and details of ligand binding are unknown. ConDockSite predictions can be tested experimentally by measuring the effects of mutagenesis of predicted ligand binding sites on ligand binding. Such efforts should be straightforward given our previous publication describing methods for recombinant expression and ligand binding assays for GPER (Souza et al., 2019).

Methods

Protein surface conservation

GPCR protein sequences were acquired from the SwissProt database (Boeckmann et al., 2005). For this study, we chose the protein sequences manually for consistency and reproducibility. For the A2A adenosine receptor, the protein sequences aligned were from Homo sapiens, Canis familiaris, Xenopus tropicalis, Myotis davidii, Loxodonta africana, Gallus gallus, Anolis caronlinesis, Oncorhynchus mykiss, Ailuropoda melanoleuca, and Alligator mississippiensis. For the β2 adrenergic receptor, the protein sequences aligned were from Homo sapiens, Oncorhynchus mykiss, Myotis brandtii, Callorhinchus milii, Ophiophagus hannah, Canis familiaris, Loxodonta africana, Ailuropoda melanoleuca, Ficedula albicollis, and Xenopus laevis. GPER protein sequences aligned were from diverse species: Homo sapiens, Rattus norvegicus, Mus musculus, Macaca mulatta, Danio rerio, and Micropogonias undulatus. Sequences were chosen to represent a diverse range of animal species. Sequences for the other receptors studied were chosen automatically by ConSurf from the NCBI “NR” non-redundant database. We obtained qualitatively similar results when using ConSurf to automatically select protein sequences for analysis. Multiple sequence alignment files were submitted to ConSurf (https://consurf.tau.ac.il) (Armon, Graur & Ben-Tal, 2001; Ashkenazy et al., 2010). ConSurf assesses conservation using Bayesian reconstruction of a phylogenetic tree. Each sequence position is scored from 0–9, where 9 indicates that the amino acid was retained in all the organisms (Fig. S2). Values from ConSurf were mapped onto the receptor surface with Chimera (Pettersen et al., 2004).

Homology modeling and docking

The crystal structures for the A2A adenosine receptor and the β2 adrenergic receptor were acquired from the RCSB protein data bank. The crystal structures used were of the β2 adrenergic receptor bound to the agonist, epinephrine (PDB 4ldo), the agonist, BI-167107 (PDB 3p0g), the inverse agonist, carazolol (PDB 2rh1), and the inverse agonist, ICI 118,551 (PDB 3ny8). The crystal structures used were of the A2A adenosine receptor bound to the agonist, adenosine (PDB 2ydo), the agonist, CGS21680 (PDB 4ug2), the inverse agonist, ZM241385 (PDB 5k2a), and the inverse agonist, compound 12X (PDB 5iub). The crystal structures of the mu opioid, serotonin 5HT2B, dopamine D2 with haloperidol, dopamine D2 with risperidone, ghrelin, histamine H1, and muscarinic M1 receptors were taken from PDB entries 5c1m, 6drz, 6luq, 6cm4, 6ko5, 3rze, and 5cxv. Structures were prepared for docking with Chimera by removing extraneous chains, solvent atoms, and bound ligands with the DockPrep protocol. Ligands were docked into receptors with SwissDock (http://www.swissdock.ch) (Grosdidier, Zoete & Michielin, 2011a).

The homology models for all the receptors were generated by I-TASSER (https://zhanglab.dcmb.med.umich.edu/I-TASSER) (Yang et al., 2015). I-TASSER generates composite homology models using multiple crystal structures. Only the predicted 7-transmembrane regions were input to I-TASSER for modeling. Models with >60% sequence identity were excluded as modeling templates so that I-TASSER would not simply retrieve the corresponding crystal structures for modeling. The PDB crystal structures used for modeling the A2A receptor were 6oij, 6e59, and 6me6. The PDB crystal structures used for modeling the β2 adrenergic receptor were 6me8, 7bts, 5zbh, 6oij, 6kuw, 6ibl, and 2ks9. The PDB crystal structures used for modeling the mu opioid receptor were 4n6h and 4djh. The PDB crystal structures used for modeling the serotonin 5HT2ϑ receptor were 6me6, 6kuw, 4iaq, 5wiv, 6a93, 5zbh, 6kp6, 7lcy, 6a94, and 6vdp. The PDB crystal structures used for modeling the dopamine D2 receptor were 6kux, 6kp6, and 6kuw. The PDB crystal structures used for modeling the ghrelin receptor were 4n6h and 6e59. The PDB crystal structures used for modeling the histamine H1 receptor were 6oij, 6kp6, 6me6, and 6kux. The PDB crystal structures used for modeling the muscarinic M1 receptor were 6me6, 6kp6, and 6me2.

The crystal structure of GPER has not yet been determined. We created a homology model using GPCR I-TASSER (Iterative Threading Assembly Refinement), the most accurate homology modeling software customized for GPCRs (https://zhanglab.dcmb.med.umich.edu/GPCR-I-TASSER) (Zhang et al., 2015). GPCR I-TASSER modeled the GPER structure using template fragments automatically selected from the closest related GPCR crystal structures (CCR5: PDB 4mbs, sphingosine 1-phosphate: PDB 3v2y, CXCR4: PDB 3odu, delta opioid: PDB 4n6h). The homology model was validated with ERRAT (Colovos & Yeates, 1993). Coordinates for E2, G1, G15, and tamoxifen were downloaded from the ZINC ligand database (Irwin et al., 2012) and submitted to SwissDock (Grosdidier, Zoete & Michielin, 2011a) for docking. SwissDock is a web interface to the EADock DSS (Grosdidier, Zoete & Michielin, 2011b) engine, which performs blind, global (does not require targeting of a particular surface) docking using the physics-based CHARMM22 force field (Brooks et al., 2009). The “FullFitness Score” calculated by SwissDock using clustering and the FACTS implicit solvent model (Haberthür & Caflisch, 2008) was used as the “Energy Score” for our calculations. SwissDock was chosen both for its high effectiveness as well as ease of use by students. For consistency, we performed all docking studies in this paper with SwissDock although we obtained qualitatively similar docking results with AutoDock Vina, the most popular docking software, in our preliminary studies.

Combined analysis

SwissDock poses were analyzed for ligand binding sites near highly conserved surfaces. Ligand binding surfaces included residues with atoms within 3.5 Å from the docked ligand. The average conservation score of the amino acids that were highlighted served as the “Conservation Score” of that specific orientation (Scheme 1). The combined ConDockSite score is defined as the product of the Conservation and Energy Scores. As the Energy Score is a modified free energy function, a highly negative ConDockSite score is associated with a more probable ligand binding site. Binding sites predicted by ConDockSite results were compared with those predicted by CASTp (http://sts.bioe.uic.edu/castp/index.html) (Dundas et al., 2006), SiteHound (Hernandez, Ghersi & Sanchez, 2009), and Concavity (Capra et al., 2009). For CASTp, SiteHound, and Concavity, ligand binding pockets were defined as residues within 4 Å of the selected probe/cluster.

Scheme 1 Calculation of combined ConDockSite scores for ligand binding sites.

The Conservation Score is calculated over the n residues in a binding site, indexed by k.

Binding site prediction benchmarks

The distances between the centers of mass of the predicted and experimental ligand coordinates served as a benchmark of comparison for the sites predicted by the ConDockSite scoring function. This is a more appropriate measure than the ligands RMSD used for pose comparison as ConDockSite is intended to predict binding sites rather than precise binding poses.

Results

We developed ConDockSite to predict ligand binding pockets using information from surface conservation and docking calculations. ConDockSite uses a simple scoring function that is the product of surface conservation scores from ConSurf (Armon, Graur & Ben-Tal, 2001) and docking scores from SwissDock (Grosdidier, Zoete & Michielin, 2011a). A highly negative ConDockSite score is associated with a more probable ligand binding site.

The A2A adenosine and β2 adrenergic receptors are by far the most heavily studied GPCRs by crystallography. First, we show that the Consurf surface conservation information alone is inadequate for identifying binding sites. Conservation varies greatly associated throughout the two receptors and is not directly associated with the ligand binding sites (Fig. S3). Many of the conserved regions are instead associated with the internal packing of the seven transmembrane helices. Next, we show that geometric binding pocket analyses, such as CASTp (Dundas et al., 2006), often work poorly for GPCRs. For the two receptors, CASTp merely predicts the entire GPCR central cavity as the ligand binding site (Fig. S4). It fails to localize the ligands to specific parts of the central cavity.

Finally, we used the crystal structures of the two receptors as positive control experiments to validate the effectiveness of ConDockSite for predicting ligand binding sites. For both receptors, we performed cross-docking of an agonist and inverse agonist against a crystal structure of the receptor bound to a different agonist or inverse agonist: ligands were cross-docked rather than self-docked into its own crystal structure. Self-docking is often trivial for modern docking methods and thus, was not studied. Docking was performed with SwissDock, which has demonstrated high accuracy in docking ligands into receptors without prior knowledge of the binding site (also known as global or blind docking), and also includes a user-friendly web interface suitable for students (Grosdidier, Zoete & Michielin, 2011a). SwissDock docking results were then ranked by the ConDockSite scoring function (Table S1). Residues within 3.5 Å of the highest scoring predicted ligand sites were compared with the binding surfaces associated with the ligand poses in the crystal structures.

As a convenient metric for the distances between predicted and experimental ligand binding sites, we use the distances between the ConDockSite-scored ligand poses and those observed in the crystal structures. For the A2A adenosine receptor, the agonist, adenosine, was cross-docked into the crystal structure of the receptor with the agonist CGS21680 (PDB 4ug2). The highest ConDockSite-ranked pose for adenosine within the A2A adenosine receptor was within 0.4 Å of the ligand position (distance between centers of mass) in the crystal structure (PDB 2ydo). (Fig. 1A). The ConDockSite-predicted binding site had a ConSurf conservation score of 0.86 and is essentially the same as the experimental binding site. The inverse agonist, ZM241385, was cross-docked into the crystal structure of the receptor with the inverse agonist, compound 12X (PDB 5iub) (Segala et al., 2016). The highest ranked site for ZM241385 within the A2A adenosine receptor was within 1.0 Å of the ligand’s position in the crystal structure (PDB 5k2a). In this top pose, ZM241385 is found within the same binding site as that observed in the crystal structure (Fig. 1B), with a ConSurf conservation score of 0.86. For both adenosine and ZM241385, the ConDockSite-predicted site corresponded to the top site predicted by SwissDock. In this case, docking alone was adequate to identify the binding site. The correct binding site also had the highest surface conservation. The A2A adenosine receptor structures are easy tests for ligand prediction, passed by both SwissDock and ConDockSite.

Figure 1 Predicted and experimental ligand binding sites in A2A adenosine and β2 adrenergic receptors.

Superposition of crystal structure with ligand bound (red) with ConDockSite predicted pose (blue). (A) Adenosine with A2A receptor. (B) ZM241385 with A2A receptor. (C) Epinephrine with β2 adrenergic receptor. (D) Carazolol with β2 adrenergic receptor.

For the β2 adrenergic receptor, the agonist, epinephrine, was cross-docked into the crystal structure of the receptor with the agonist, BI-167107 (PDB 3p0g) (Rasmussen et al., 2011). The highest ranked pose for epinephrine within the β2 adrenergic receptor was within 0.4 Å of the ligand position within the crystal structure (PDB 4ldo). This binding site for epinephrine was again essentially the same as the observed binding pocket (Fig. 1C) and had a ConSurf conservation score of 0.85. This site was also scored the highest by SwissDock. For the β2 adrenergic receptor, the inverse agonist, carazolol, was cross-docked into the crystal structure of the receptor with the inverse agonist ICI 118,551 (PDB 3ny8) (Wacker et al., 2010). The highest ranked pose for carazolol within the β2 adrenergic receptor was within 1.0 Å of the ligand’s position within the crystal structure (PDB 2rh1). This binding site for carazolol was essentially the same as that in the crystal structure (Fig. 1D). This binding pocket has a ConSurf conservation score of 0.78. The correct carazalol binding site was scored the third highest by SwissDock. The use of surface conservation information allowed selection of the proper binding site. The extremely accurate placement of both agonists and inverse agonists for the A2A adenosine and the β2 adrenergic receptors demonstrates ConDockSite’s effectiveness when crystal structures are available.

Unfortunately, crystal structures are not available for most GPCRs. The most valuable use of ConDockSite is to predict drug binding sites in homology models. By using surface conservation information, ConDockSite is less sensitive to homology model inaccuracies than other ligand binding site prediction methods that are based purely on geometric methods. To demonstrate the ability of ConDockSite to work with homology models, we created models of eight GPCRs, the β2 adrenergic, A2A adenosine, 5HT2B serotonin, mu opioid receptors, D2 dopamine, ghrelin, H1 histamine, and M1 muscarinic receptors, while excluding their known X-ray structures from the templates used for modeling. We used I-TASSER (Yang et al., 2015) for homology modeling which does not use GPCR-specific structural constraints but allows for custom selection of templates. I-TASSER created fairly accurate models of all eight receptors, with RMSDs across Cα atoms between the models and crystal structures ranging from a best of 1.5 Å for the histamine H1 receptor (PDB 3rze) to a respectable 2.1 Å for the muscarinic M1 (PDB 5cxv) receptors. We used ConDockSite to predict the binding sites of the β2 adrenergic receptor with carazalol, A2A adenosine receptor with ZM241385, 5HT2B serotonin receptor with methysergide, mu opioid receptor with BU72, D2 dopamine receptor with risperidone and haloperidol, ghrelin receptor with the antagonist Compound 21, histamine H1 receptor with doxepin, and muscarinic M1 receptor with tiotropium. ConDockSite performed best with the β2 adrenergic receptor homology model, with only 2.3 Å between the centers of mass of the predicted and crystal structure ligand poses (Fig. 2, Table S2), supporting the prediction of very similar binding pockets. The binding pocket was predicted correctly although the ligand pose was inaccurate. There was a weak correspondence between ConDockSite performance with the accuracy of the homology models. The H1 histamine receptor model was the best homology model, with RMSD of 1.5 Å between the model and the crystal structure Cα atoms. However, ConDockSite was unable to correctly predict the binding site for the antagonist doxepin, which was predicted >10 Å from the crystal structure site. The poor performance was due to the inaccurate modeling of a loop containing residues 162–165 over the top of the ligand binding site as well as multiple errors in the conformation of side chains in the transmembrane helixes. The D2 dopamine receptor model was the second-best homology model, with RMSD of 1.6–1.8 Å between the model and the crystal structure Cα atoms (PDB 6luq bound to haloperidol and 6cm4 bound to risperidone). ConDockSite predicted the binding site of risperidone well, only 2.4 Å from the crystal structure site, but less well for haloperidol, which was predicted 4.3 Å from the crystal structure site. With this medium level of accuracy, the ligand poses are generally incorrect, but most of the binding residues in the predicted pockets are the same as those in the crystal structures, supporting successful ConDockSite predictions. The ghrelin receptor homology model was also predicted well with RMSD of 1.7 Å between the model and the crystal structure Cα atoms (PDB 6ko5). The ghrelin antagonist, Compound 21, was predicted 3.1 Å away from the crystal structure site. The predicted ligand binding sites for the A2A adenosine and mu opioid ligands (PDB 5c1m) are 3–4 Å away from the crystal structures. ConDockSite performs less well with the serotonin 5HT2B receptor (PDB 6drz) where the distance between the predicted and actual ligand binding sites was 7.0 Å. In the serotonin 5HT2B receptor structure, the ligand, methysergide, binds very deep in the receptor. Serious errors in homology modeling of the 5HT2B receptor side chains make it difficult or impossible to dock the ligand into the deep, restricted binding site. ConDockSite also fails with the muscarinic M1 receptor (PDB 5cxv) where the distance between the predicted and actual triotropium binding sites was >10 Å. Overall, our results with ConDockSite are consistent with benchmark modeling results that show that GPCR homology models of modest accuracy from templates with low sequence identity are still sometimes useful for docking and virtual screening (Lim et al., 2018; Costanzi et al., 2019). In some of the failed cases (histamine H1, muscarinic M1, and serotonin 5HT2B), while the homology models were sometime accurate at the level of backbone atoms in the 7tm region, the loops were modeled poorly and disrupted the modeled ligand binding pocket. In other cases, the homology model backbones were modeled well but differences in the side chain conformations disrupted the integrity of the ligand binding sites. In these cases, the homology models are not accurate enough for docking or ConDockSite. In retrospect, ConDockSite sometimes performed better with alternative homology models generated by I-TASSER with lower RMSD measures but with alternative loop placements.

Figure 2 Predicted and experimental ligand binding sites for homology models of eight GPCRs.

Models are shown in order of best to worst predictions. Superposition of crystal structure with ligand bound (red) with ConDockSite predicted pose (blue). (A) Carazolol with β2 adrenergic receptor. (B) Risperidone with dopamine D2. (C) Compound 21 with ghrelin receptor. (D) BU72 with mu opioid. (E) Haloperidol with dopamine D2. (F) ZM241385 with A2A adenosine. (G) Methysergide with serotonin 5HT2B. (H) Doxepin with histamine H1. (I) Tiotropium with muscarinic M1.

After demonstrating the applicability of ConDockSite for homology models, we applied ConDockSite to predict the binding sites in a GPCR, GPER (G-protein coupled estrogen receptor), which has not yet been crystallized. GPER is of great interest to us and other researchers due to its unusual physiological and pharmacological roles in estrogen-related biology. As there is no experimental data on the ligand binding sites in GPER, we cannot validate our predictions. They should be considered speculative at this point, but we hope they can guide future experiments. To predict the potential ligand binding sites in GPER, we first created a homology model using GPCR-I-TASSER (Zhang et al., 2015). GPCR-I-TASSER has been shown to be among the most accurate GPCR homology modeling software package. We used the generic I-TASSER in our validation studies because of its fine-grained options for template selection that are lacking in GPCR-I-TASSER. Because GPCR-I-TASSER uses GPCR-specific structural constraints, it is expected to outperform the generic I-TASSER (Zhang et al., 2015). GPCR-I-TASSER identified the closest matching crystal structure to GPER to be the CCR5 chemokine receptor (PDB 4mbs) with 23% sequence identity. GPCR-I-TASSER used this crystal structure along with 9 other GPCR structures as templates for homology modeling. The GPER homology model differs from chain A of the crystal structure of CCR5 chemokine receptor with RMSD of 0.96 Å across Cα atoms (Fig. S5) and has a Ramachandran plot with 95.6% of residues, excluding glycine, in preferred regions (Fig. S6). The primary differences are in the extracellular loop between helices 4 and 5 (ECL2) and the intracellular loops between helices 5 and 6 (ICL3), and after helix 7. These two intracellular loops are predicted by ERRAT (Colovos & Yeates, 1993) to be the least reliable based on the likelihood of atom pair type interactions from high-resolution crystal structures (Fig. S7).

Using the SwissDock server (Grosdidier, Zoete & Michielin, 2011a), we docked structures of the four ligands E2, G1, G15, and tamoxifen (Fig. S1) to the homology model of GPER. The docked sites from SwissDock, including those that were scored the highest, were located throughout the receptor surface and thus were considered mostly nonviable (Fig. 3). The shortcomings of a purely physics-based scoring function such as that used by SwissDock in predicting ligand binding are not surprising given the lack of an experimental crystal structure and well-known limitations of current homology modeling and docking methodology (Li, Hou & Goddard, 2010; Merz, 2010; Wan et al., 2015; Smith et al., 2016).

Figure 3 E2 binding sites calculated by SwissDock.

E2 poses are in blue. The top of the figure corresponds to the extracellular face of GPER.

We then ranked all ligand binding sites generated by SwissDock using the combined ConDockSite score. For all four ligands, the ConDockSite score identified one or two ligand binding sites that clearly outscored (more negative) other candidates (Table S1). ConDockSite identified the same approximate binding site for all four ligands, although this was not an explicit criterion in the calculations (Fig. S8). The average ConSurf conservation score across the four ligand binding sites is 0.82 (1.0 represents complete conservation), indicating that the site is highly but not completely conserved. The binding site is located deep in the receptor cleft, although depth was not a criterion in the prediction calculation. Given the lack of additional experimental evidence for the location of the ligand binding site, the proposed ConDockSite sites are physically reasonable.

We found two promising potential binding sites for E2 in GPER. The two sites are 4.4 Å apart, located deep in the receptor cleft (Fig. 4). E2 is oriented perpendicular to the lipid membrane and rotated about 180° between the two poses. The conservation scores for these two poses are 0.84 and 0.80. The energy scores of the two poses are similar. The amino acids contacting E2 in pose 1 are conserved in GPERs from six species, and only one residue contacting pose 2, H282, varies across species. In the top ranked pose, there is a hydrogen bond between the inward pointing D-ring hydroxyl group of E2 and the carboxyl terminal on E115. Hydrophobic interactions are present between E2 and non-polar residues L119, Y123, P303, and F314. This binding site approximately corresponds to that predicted by Lappano et al using docking (Lappano et al., 2010). In the second ranked pose, the inward pointing A-ring hydroxyl group of E2 makes a hydrogen bond with N310. This pose is in a less hydrophobic environment, contacting primarily H282 and P303.

Figure 4 Predicted E2 binding sites in GPER.

(A) The two highest scoring docking poses for E2. (B) Receptor-ligand interactions for E2 pose 1. (C) Receptor-ligand interactions for E2 pose 2.

ConDockSite predicts that G1 and G15 bind in adjacent but distinct binding sites separated by 2.3 Å. The top predicted binding site for G1 is found within the pocket bound by Y55, L119, F206, Q215, I279, P303, H307, and N310 (Fig. 5). This orientation had the highest conservation score of all predicted binding sites at 0.85. In this pose, N310 makes a long hydrogen bond (3.6 Å N-O distance) with the acetyl oxygen of G1. The predicted binding site for G15 is found within the pocket is surrounded by L119, Y123, M133, S134, L137, Q138, P192, V196, F206, C207, F208, A209, V214, E218, H307, and N310. This pose had a conservation score of 0.8. Hydrogen bonding is not observed between GPER and G15. Hydrophobic interactions are observed with L119, Y123, F206, and V214. The ConDockSite G1 result correspond to the binding sites predicted by recent studies using docking and molecular dynamics simulations and validated by design and activity testing of new G1 derivatives (Méndez-Luna, Bello & Correa-Basurto, 2016; Martínez-Muñoz et al., 2018).

Figure 5 Predicted G1 and G15 binding sites in GPER.

(A) The highest scoring docking poses for G1 (maroon) and G15 (cyan). (B) Receptor-ligand interactions for G1. (C) Receptor-ligand interactions for G15.

ConDockSite predicted two equally high scoring, overlapping poses for tamoxifen, near E115, L119, Y123, L137, Q138, M141, Y142, Q215, E218, W272, E275, I279, P303, G306, H307, and N310 (Fig. 6). The conservation score of this orientation is 0.81. Hydrophobic interactions are observed between tamoxifen and non-polar residues L119, Y123, Y142, P303, and F314. Notably, the amine group of tamoxifen makes ionic interactions with E218 and E275.

Figure 6 Predicted tamoxifen binding sites in GPER.

The highest scoring docking poses for tamoxifen, pose 1 (maroon) and pose 2 (cyan).

We compared the GPER ligand binding sites predicted by ConDockSite to those predicted by three other software packages representing different approaches: CASTp (Dundas et al., 2006), which analyzes surface geometry, SiteHound (Hernandez, Ghersi & Sanchez, 2009), which maps surfaces with a chemical probe, and Concavity (Capra et al., 2009), which analyzes surface geometry and conservation (Fig. 7). All three methods could identify a ligand binding site very roughly matching that from ConDockSite. In comparison with the ligand binding sites predicted by traditional methods based on surface geometry and conservation (Fig. 7), the sites predicted by ConDockSite are more detailed in shape due to the information from chemical interactions from ligand docking. The pocket predicted by ConDockSite is deeper than the other pockets, which while intuitively attractive, is not necessarily correct. SiteHound performed particularly poorly, with the top scoring site located on the GPER intracellular face. The site identified by SiteHound closest to the ConDockSite site was scored third and is a shallow binding pocket near H52-G58, E275-H282, and R299-H307 (Fig. 7C). In contrast, the Concavity site was smaller and shallower than the ConDockSite site (Fig. 7D). Surprisingly, the site predicted by the simpler CASTp method best matched the ConDockSite site but is also smaller and shallower (Fig. 7B). For proteins such as GPCRs with large, concave binding pockets, geometry-based prediction methods such as Concavity and CASTp can easily identify the general, approximate location of the ligand binding site. However, such methods may have more difficulty recovering the specific, ligand-specific binding site. It is also surprising that ConDockSite more closely matched the results of the geometry-based methods given that ConDockSite does not take surface geometry into account. As described previously, the G1 and G15 binding sites predicted by ConDockSite more closely match those made using docking against very computationally expensive molecular dynamics simulations (Méndez-Luna, Bello & Correa-Basurto, 2016).

Figure 7 Predicted E2 binding sites by ConDockSite, CASTp, SiteHound, concavity.

Ligand binding sites are colored, predicted by (A) ConDockSite. (B) CASTp. (C) SiteHound. (D) Concavity.

Discussion

The ConDockSite scoring method, incorporating information from both surface conservation and docking binding energy, demonstrated high accuracy in predicting ligand binding sites from the crystal structures of two class A GPCRs, the A2A adenosine and β2 adrenergic receptors. ConDockSite also successfully predicted the ligand binding sites for many high-quality homology models but failed for models for which loops were modeled poorly and disrupted the ligand binding site. Better homology models with improved loop and side chain placement should allow better ConDockSite performance. ConDockSite was also used to predict viable ligand binding sites for four different GPER ligands. In contrast to more typical geometry-based ligand binding site prediction methods, ConDockSite scoring takes advantage of chemistry-specific information about the ligand-receptor interface. The poor performance of SiteHound in predicting ligand binding sites on GPER suggests that a method based only on chemical interactions or docking is highly susceptible to error, most likely due to the inadequate accuracy of homology models. Surface conservation data not only provides orthogonal knowledge but also dampens the influence from the shortcomings of current computational methods in homology modeling, docking, and predicting binding affinity. How best to mathematically combine these multiple data sources has been debated (Capra & Singh, 2007; Capra et al., 2009), but we demonstrate here that a simple product scoring function is already effective. The four GPER ligands studied here differ greatly in chemical structure, but the ConDockSite scoring method predicted that all four bind to the same approximate region, deep in the extracellular cleft of the receptor. Undoubtedly, further refinement of a hybrid scoring function will lead to improved predictions.

Earlier GPER modeling studies using molecular dynamics simulations and docking identified different potential binding sites for E2, G1, and G15 near F206 and F208; the interaction with this region was described as driven primarily by π–π stacking interactions (Arnatt & Zhang, 2013; Méndez-Luna et al., 2015). Figure S9 compares the ConDockSite binding site against that predicted in the molecular dynamics simulation and docking study. The ConDockSite binding site is located deeper in the extracellular cleft; the other proposed site involved more surface-exposed loops. It was proposed that Q53, Q54, G58, C205, and H282 all interact with G1 and G15; however, none of these residues are conserved across the six species we analyzed. More recent studies using better homology models and computationally expensive long time-scale molecular dynamics simulations predict E2, G1, and G15 binding sites that approximately match those predicted by ConDockSite (Lappano et al., 2010; Méndez-Luna, Bello & Correa-Basurto, 2016). The ConDockSite binding site predictions can be tested experimentally by performing site-directed mutagenesis and ligand binding assays.

In summary, the simple ConDockSite hybrid scoring model predicts physically plausible ligand binding sites by combining information from ligand docking and surface conservation. Using multiple orthogonal sources of information partially avoids errors introduced by modeling (Capra et al., 2009). Given a homology model of modest quality, ConDockSite can sometimes accurately predict ligand binding sites. Using this hybrid method, we identified a site in the extracellular cavity of GPER that has the potential to bind four known GPER ligands. Further optimization of hybrid scoring functions and homology modeling methods should yield significantly improved predictions. Extension of this approach may allow analysis of non-class A GPCRs.

Supplemental Information

Supplemental Information 1 Cross-docked and homology models.

Click here for additional data file.

Supplemental Information 2 Supplemental Figures and Tables.

Click here for additional data file.

Additional Information and Declarations

Competing Interests

Author Contributions

Data Availability

Ho Leung Ng. is an Academic Editor for PeerJ.

Ashley Ryan Vidad conceived and designed the experiments, performed the experiments, analyzed the data, prepared figures and/or tables, authored or reviewed drafts of the paper, and approved the final draft.

Stephen Macaspac conceived and designed the experiments, performed the experiments, analyzed the data, prepared figures and/or tables, authored or reviewed drafts of the paper, and approved the final draft.

Ho Leung Ng. conceived and designed the experiments, performed the experiments, analyzed the data, prepared figures and/or tables, authored or reviewed drafts of the paper, and approved the final draft.

The following information was supplied regarding data availability:

Our analysis was performed using structures and software tools that are publicly available in databases (PDB) or web servers (I-TASSER, ConSurf, SwissDock).

Web servers:

ConSurf: https://consurf.tau.ac.il

I-TASSER: https://zhanglab.dcmb.med.umich.edu/I-TASSER

GPCR-I-TASSER: https://zhanglab.dcmb.med.umich.edu/GPCR-I-TASSER/

SwissDock: http://www.swissdock.ch

CASTp: http://sts.bioe.uic.edu/castp/index.html

PDB files:

beta2 adrenergic receptor: 4ldo, 3p0g, 2rh1, 3ny8

A2A adenosine receptor: 2ydo, 4ug2, 5k2a, 5iub

mu opioid receptor: 5c1m

serotonin 5HT2B receptor: 6drz

dopamine D2 receptor: 6luq, 6cm4

ghrelin receptor: 6ko5

histamine H1 receptor: 3rze

muscarinic M1 receptor: 5cxv.

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
