# Peer review of "Locating ligand binding sites in G-protein coupled receptors using combined information from docking and sequence conservation"

_PeerJ, doi:10.7717/peerj.12219_

## Round 0.1 · original submission · Major Revisions

We have received three very detailed reports, with significant requests for clarification and additional information. Please address them thoroughly. I also have a couple of review-level comments of my own:

1) this method combines a ConSurf score with a docking score, but we are not given any information regarding the gains in performance (over the use of only ConSurf or only docking scores) achieved by this combination. It would be helpful for the reader to see how its performance compares with the separate use of each of these scores only.

2) The paper states that " Crystal structures of receptors were screened for residues within 3.5 Å of their respective ligands. These residues served as a benchmark of comparison for the sites predicted by the ConDockSite scoring function. " I am afraid that your choice of residues renders your results are less convincing than they would be if ConDockSite had been applied to the full surface to confirm that no other region afforded a better score than the experimentally confirmed one.

Reviewer 1 ·

Basic reporting

In this study the authors have presented ConDockSite, a method that combines docking scores and surface conservation to identify the ligand binding sites. The method is then validated using two GPCR targets with known crystal structures and demonstrated using another GPCR with no known crystal structure. Overall, the manuscript is relatively well-written and the results are clearly presented. However, the Materials and Methods should be placed after the Introduction and not at the end of the manuscript.

Experimental design

In terms of experimental design, I have a few concerns as below:

• The choice of two class A GPCR structures to validate the method significantly limits any ability to truly evaluate how the method performs, since class A GPCRs have relatively well-defined binding sites within the TM bundle. While it is true that GPCRs are important drug targets, binding site prediction (the problem the method is intended to solve) is not one limited to GPCRs. The authors should have either use a broader range of targets, or clearly state the limited context in which their results are should be interpreted.
• In Line 354 it is stated that SwissDock poses were manually screened for binding sites located on or near the extracellular site of the protein. This manual intervention does not allow for an objective assessment of ConDockSite’s performance since it introduces a bias (i.e. without this manual intervention would the same results have been achieved?)

Validity of the findings

• The results obtained for the validation targets (β2 and A2A) would be highly dependent on SwissDock results. The results of the COM distance between the predicted pose and the crystal structure pose are reported, but the results obtained from only using SwissDock (without using the conservation scores) should be presented as well in order to allow readers to gauge the degree of improvement afforded by ConDockSite.
• Similarly, for these two targets some comparison with the results of other binding site prediction programs would also be useful, since this is done for GPER but in the case of these two targets the crystal structure is available for comparison. This could be done by either describing the binding site residues identified or using their COM distance. One would expect that for binding site prediction programs the number of binding site residues identified would be larger compared to any approach that utilizes docking as part of the method, although this is somewhat dependent on the size of the ligand as well.
• Given that the authors are presenting a new method, demonstrating the use of the method using a target GPER does not add any value, as there is no way to validate the predictions. Some comparisons with other binding site prediction tools and modelling studies are provided, but these comparisons would inherently not be precise enough to demonstrate that the results are valid. Choosing other targets with experimental data available and discussing it within that context would have been more valuable. If GPER is a target of particular interest to the authors, then some discussion in the context of any available experimental data (e.g. mutagenesis studies) would be useful. If these are not available, then the results are very speculative and should be more clearly indicated as such.
• Line 245 – it is stated that all three methods could identify a binding site roughly similar to ConDockSite. This should be elaborated more clearly in terms of the residues involved in the binding site or the size of the binding site so a proper comparison can be made. The figure provided only allows for a very qualitative assessment of this.

Other minor comments:

• Line 153 – What is the RMSD between homology models and crystal structures calculated based on? Backbone atoms? Alpha carbons?
• Line 162 – the RMSD here is referring to RMSD between ligands correct? Wasn’t the results supposed to be presented as the COM distance between ligands?
• Line 185-186 – these loops are more typically referred to as extracellular loop 2 (ECL2) and intracellular loop 3 (ICL3)
• Line 199-201 – This is a repeat from the start of the Results and is also stated in Methods.

Reviewer 2 ·

Basic reporting

The work focus on the identification of binding sites for the GPCRs superfamily. In fact, GPCRs are one of the most import drug targets because of its implication in a huge variety of diseases and therefore I consider the work relevant for the field. Furthermore, the manuscript is very well structured and written .
In my opinion, the figures could be improved, simple changes should be done, indicating, for example, the typical numenclature for the GPCRs, and the extracellular and intercellular loops and regions.
The S.I. Fig.7 should be improved, the colors used impede to see the information of the alignment.

Experimental design

The methods strategy and workflow are appropriate.
To benchmark the binding sites found with ConDocSite the authors used receptors which X-ray structures are known and have been very well characterized, such as the Adenosine A2A. Homology models were also used to validate the methodology.

Validity of the findings

I would like the authors to address the threshold that can be used to use the method with high confidence as well as the ability of ConDocSite to distinguish between agonists, partial agonists and antagonists among the different classes of GPCRs. Can the scores be correlated with the affinity of the ligands to the receptors?
Surprisingly, the scores indicated on table two, and below on lines 3 and 5, are exactly the same for the two receptors. Can the authors justify for these results?
Table S1. ConDock predicted ligand binding sites for A2A adenosine and 2 adrenergic receptors using crystal structures.

GPCR Ligand Distance to crystal structure pose (Å) ConSurf conservation score SwissDock energy Score ConDock score
A2A adenosine adenosine (agonist) 1.8 0.86 -1702.1 -1458.9
A2A adenosine ZM241385 (inverse agonist) 1.0 0.86 -1534.7 -1319.8
2 adrenergic epinephrine (agonist) 0.4 0.87 -1112.0 -970.7
2 adrenergic carazolol (inverse agonist) 1.8 0.78 -1534.7 -1319.8


The distance between ConDock and crystal structure poses presented on S.I. table 2 are very large.

Reviewer 3 ·

Basic reporting

- Line 49: according to GPCRdb the experimental structure has been solved for 70 class A GPCRs, 10 class B, 4 class C and 3 class F.
- Line 92: I suggest characterizing the binding sites as hypothetical.
- Line 126: what does the author mean with “proxy”?
- Lines 124-125 and 126-128 tell almost the same
- Line 150: rephrase with something like “by excluding their known X-ray structures from the templates”
- Lines 163-164: please, reformulate because the sentence is not clear.
- Line 177: repetition of “we used”, delete the second one
- Lines 199-202: better to move the explanation of ConDockSite at the beginning of the Results section, where the technique is introduced for the first time.
- Line 228: replace “bound by” with “surrounded by”
- Line 239: replace “is neutralized” with “makes ionic interactions with”
- Line 332: substitute “prep” with “prepared”
- Lines 333-337: move those lines referring to SwissDock at the end of the paragraph, around line 346, where SwissDock is explained in more detail.

Experimental design

- Lines 101-102: “We discuss how the ConDockSite-predicted binding sites provide a basis for G1 and G15 binding specificity” this statement is misleading. The research question about binding specificity is not addressed and answered in the manuscript.
- Lines 122-123: I am not able to find in the results the mentioned comparison between the residues within 3.5 Å of the top scoring poses and the crystal structures.
- I understand that the purpose of the project is not the prediction of accurate binding modes, but the manuscript would benefit by mentioning the RMSD between the predicted poses and the crystallographic ones.
- Line 137: the authors mention the PDB ID of one of the employed structures (4LDO) just for beta2. I suggest including the PDB IDs for all the structures that were used in the manuscript.
- Lines 136-138: I suggest including in the text the Consurf score for epinephrine as done for the other ligands.
- Line 152: which templates were used by I-TASSER to build the models? Further details about A2A, beta2, mu opioid and 5HT2B models generation are lacking in the Experimental section.
- Line 153: which are the atoms used to compute RMSDs between crystal structures and models? Ca carbon atoms? Backbone atoms? Heavy atoms?
- Line 154-155: why were 2RH1 and 5K2A used as reference to compare beta2 and A2A AR models? In the case of 2RH1 for beta2, resolution could be the reason, but I cannot understand the choice of 5K2A for A2A AR. Please explain, and in case provide RMSD to some more valuable experimental structure.
- Line 162: the authors mention a “RMSDs of 3-4 Å between the predicted and crystal structure ligand poses”, but in table 2 the 3-4 Å is related to the distance between the ligands’ centers of mass. The authors should correct this incongruence, and, in addition, I suggest adding the ligands RMSD in addition to centers of mass distance, also in this case.
- Line 163: “in this RMSD range”: what do the authors mean? And again, is this RMSD or centers of mass distance?
- Line 165-166: again, is this RMSD or distance?
- How did the authors choose the sequences for the alignment and ConSurf calculation?
- How did the authors generate the multiple sequence alignment? Why just ortholog genes were employed?
- Line 182: which are the other 9 GPCRs structures employed by GPCR-I-TASSER? How were they used?
- Line 184: avoid the use of “excellent” to describe a result. The authors should instead mention how the models’ phi and psi dihedral angles fit the favored regions of the Ramachandran plot, and/or how many outliers are present.
- Line 203: can the authors explain what they mean with “outscored”? Which cutoff was employed?
- Line 198: does “we then ranked all ligand binding sites” mean that the poses located throughout the receptor and considered “nonviable” (line 193) were considered in the evaluation? Did they present a low ConDockSite score enabling the discrimination from the poses in the putative orthosteric site? Reading line 354 it seems that poses far from the extracellular site of the receptor have been manually removed, and this would distort the accuracy of the method, given that its aim should be the effective prediction of binding sites.
- Lines 226-227: the hydrogen bond with Asn310 is not visible from the image, please, remake the image to make in clear. Moreover, what does “long hydrogen bond” mean? Long?

Validity of the findings

- According to what reported in the paper DOI: 10.1038/NCHeMBIO.2266, “phylogenetically unrelated GPCRs frequently share ligands” and “homology-related receptors frequently differ substantially in their ligand selectivity”. Moreover, looking at the multiple sequence alignment, a high percentage of residues show high conservations, so I would expect a leveling of the ConSurf scores, because it does not seem rare for a ligand to interact with highly conserved residues. In other terms, I think that the major shortcoming of this work is that there is no proof of the advantage in merging SwissScore and ConSurf, because a comparison with these two single scores is lacking. I suggest, as a major revision, to include this kind of analysis as a control, and compare the pose rankings obtained with SwissScore, ConSurf and ConDockSite. Which poses would have been selected for the test cases (crystallographic A2A and β2) if just the SwissScore or the ConSurf score were used? Without this control, the advantage in using the product of the two scores, and the sentence “Incorporating sequence conservation information in ConDockSite overcomes errors introduced from physics-based scoring functions and homology modeling.” at the end of the abstract, are not supported. The only comparison with different methods has been provided in the case of GPER, but there is no experimental data making this test case a true positive, and, in addition, among the methods that have been compared there is no pure SwissScore nor ConSurf.
- Lines 167-169: which are the “serious errors in homology modeling” the authors refer to? The RMSD between model and template reported in Table S2 is just 1.8Å for 5HT2B receptor, comparable or even lower than more successful cases like mu opioid and A2A AR. How can the authors explain the 7.3Å distance (or RMSD) between predicted and crystallographic ligand? Which are the “serious errors” in the model? Maybe the RMSD between model and crystal structure that was employed is not sufficient to explain 5HT2B model limits? Any thoughts? Or is the depth of the binding pocket the problem? In that case, it would be better to reformulate the sentence.
- Why did the authors employ beta2 and A2A AR for the cross-docking validation, and add mu opioid and 5HT2B just for the homology modeling validation stage? Given the limits of mu opioid and especially 5HT2B models, the authors should add these receptors to the cross-docking validation stage. In this way it will be clear if the problems related to mu opioid and especially 5HT2B are model-related, or instead receptor/binding pocket-related.
- Lines 169-171: “Nevertheless, our results with ConDockSite are consistent with benchmark modeling results that show that GPCR homology models of modest accuracy from templates with low sequence identity are still useful for docking and virtual screening” How can the authors state this? The authors have just shown that the 5HT2B model is for some reason poor (even if it is not clear why) and do not provide good docking results, so how can it be useful for docking and VS? I see some discrepancy between what just observed by the authors and what stated citing the literature.
- Lines 245-248: what does “more detailed and of higher resolution due to the information from chemical interactions from ligand docking” mean? I cannot get what details and resolution are in this context.
- Lines 248-249: I also cannot understand this point. What does it mean that the other methods are not able to differentiate among sites for different ligands? This statement is not supported by any observation/result.
- Lines 291-292: from Figure S6 it does not seem that “the other proposed site mostly involved surface-exposed loops”

---

## Round 0.2 · Minor Revisions

I am generally satisfied with your responses, but there are still a few places where I think some language tweaking/clarification is in order:

A) "SwissDock poses were manually screened for those binding sites located on or near the extracellular side of the protein. We performed manual screening as a quick way to assess the output. It did not affect the accuracy or bias of our results as these external sites always scored very poorly using the ConDockSite method due to poor conservation of the binding sites. Manual screening can be omitted without affecting the results." This description seems quite contradictory: if the external sites always scored poorly and had to be identified manually, how can the manual screening step be ommited without changing the results? Later in the paper, authors do state that extracellular binding-sites are expected to be spurious, but at this point in the paper no such mention has been made, nor has any rational for screening for binding sites on that region been presented, and therefore the text is quite jarring. Please clarify all this.


B) in the "Binding site prediction benchmarks" section, the sentence "This is a more appropriate measure than the ligands RMSD used for pose comparison as ConDockSite is intended to predict binding sites." should probably be changed to "This is a more appropriate measure than the ligands RMSD used for pose comparison as ConDockSite is intended to predict binding sites rather than precise binding poses"
C) in the beginning of results section "First, we show that the Consurf surface conservation information is inadequate for identifying binding sites. " should probably be changed to "First, we show that the Consurf surface conservation information ALONE is inadequate for identifying binding sites. "

D) I tend to agree with reviewer #2's request of a larger benchmark, but I do not know if many other structures of GPCR-ligand complexes are available. Testing of your method against a few more examples would be highly desirable and (in my opinion) considerably aid in the adoption of your method by the community, but I will not make that a requirement for acceptance.

Reviewer 1 ·

Basic reporting

The manuscript is well-presented.

Experimental design

The authors have not made any changes to the initial experimental design, which potentially limits the applicability of the method to Class A GPCRs. However, this is now more appropriately acknowledged in the manuscript, allowing readers to consider the findings in context.

Validity of the findings

The authors have added some results and discussion that adequately address the previously-raised concerns.

Additional comments

The authors have improved the manuscript based on the previous comments. While some limitations do remain, the manuscript now acknowledges them more readily and provides better context for the potential domain of applicabilty for this method.

Reviewer 2 ·

Basic reporting

The article does not address all the questions asked and, in my opinion, the authors claim results that are not fully proven.

Experimental design

Techniques are ok but the pipeline could be improved.

Validity of the findings

In my opinion the results are not robust and it needs to be benchmarked.

---

## Round 0.3 · Minor Revisions

I am quite satisfied with the latest version. I have one small request, though: you state "In all the failed cases (histamine H1, muscarinic M1, and serotonin 5HT2B), while the homology models were sometime accurate at the level of backbone atoms in the 7tm region, the loops were modeled poorly and disrupted the modeled ligand binding pocket. " , but do not show a picture showing that the worst-performing models have worse loop-modeling than the others. Please provide a figure with those comparisons, and add the model coordinates to the suppporting information

---

## Round 0.4 · Minor Revisions

I am afraid that the models you submitted have been wrongly chosen, since they do not agree with the description in the paper. Please see the attached PDF and correct these issues. There also does not seem to be a consistent relationship between the quality of the loop modeling and the correctness of the docking pose, which suggests that changes must be introduced in the text in lines 313-317 ( which currently reads "In all the failed cases (histamine H1, muscarinic M1, and serotonin 5HT2B), while the homology models were sometime accurate at the level of backbone atoms in the 7tm region, the loops were modeled poorly and disrupted the modeled ligand binding pocket. In these cases, the homology models are not accurate enough for docking or ConDockSite")

---

## Round 0.5 · accepted · Accept

Thank you for the clarifications. I am glad to accept you manuscript for publication in PeerJ!

---

## Author Rebuttal · Round 0.5

Dear Prof. Pedro Silva,

I thank you for your careful reading of our manuscript and your constructive feedback.

The statistics and text have been corrected to include the correct values. The overall conclusions of the study are not affected. I have also uploaded the homology model coordinates as supplementary materials as requested.

Ho Leung Ng

8/28/2021

# Editor comments (Pedro Silva)

MINOR REVISIONS

Editor comments in red. My response in blue.

A) the loops in the beta2 model (171-196) are quite different from the one in the crystal structures (3p0g, 4ldo), and (more importantly) they keep the entrance to the inner channel more open than in those crystal structure). Also, the beta2 model provided contains carazolol, but the pose is quite different (RMSD=5.9 angstrom) from the one in structure 2rh1, incorntrast to the very good fit claimed in lines 257-258. Since your method does not prevent it from finding good docking poses in beta2, the disappointing behavior you found in H1, M1 and 5HT2B models cannot simply be attributed to a bad loop model. For this reason, I would suggest rephrasing the text in lines 313-317 ("In all the failed cases (histamine H1, muscarinic M1, and serotonin 5HT2B), while the homology models were sometime accurate at the level of backbone atoms in the 7tm region, the loops were modeled poorly and disrupted the modeled ligand binding pocket. In these cases, the homology models are not accurate enough for docking or ConDockSite") .

Lines 257-258 refer to the comparison of the cross-docking results rather than homology models.

We agree that our descriptions of loop differences were too simplistic. We have changed lines 313-317 to read, "In some of the failed cases (histamine H1, muscarinic M1, and serotonin 5HT2B), while the homology models were sometime accurate at the level of backbone atoms in the 7tm region, the loops were modeled poorly and disrupted the modeled ligand binding pocket. In other cases, the homology model backbones were modeled well but differences in the side chain conformations disrupted the integrity of the ligand binding sites."

B) The loops in the A2A model (143-166) are also quite different from the ones in the crystal structures (2ydo, 4ug2,5iub,5k2, for A2A) : in this case, the modelled structure is more open than the crystal structures, and similar to that in structure PDB:5c1m of the mu-opioid receptor which leads me to believe that the differences in loop structure you find may suimply reflect different

physiological conformations of the GPCR. Regardless of the origin of those differences, neither the A2A (or the beta2 models) deposited as SI contain adenosine (or adrenaline), but a different molecule in quite a different pose. This makes it impossible to reproduce Fig. 1A and 1C.

Lines 143-166 and Figs. 1A and 1C refer to results from ConDockSite on cross-docked crystal structures rather than homology models. I have changed the Figure 1 heading to "Predicted and experimental ligand binding sites in A2A adenosine and beta2 adrenergic receptors from cross-docked crystal structures" to make this clearer. I have also added the cross-docked models to the SI.

C) Furthermore, in the 5HT2 model, the modelled loops (corresponding to aa 194-206 in DRZ) that are different from the ones in the 6drz structure are away from the binding site entrance, and in the H1 model, the loop actually leaves the entrance to the binding site more unencumbered than in the 3rze model. It therefore does not seem at all appropriate to attribute the observed poor performance of ConDurfDock in these instances to this loop (especially in comparison to the good behavior in spite of poor loops described above).

In the 5ht2b model, the problem loop in the homology model is actually aa 167-170, which extends Leu 169 into the correct ligand binding site. But there are also problems in the modeled side chain conformations which we described in lines 309-311: "Serious errors in homology modeling of the 5HT2B receptor side chains make it difficult or impossible to dock the ligand into the deep, restricted binding site." For the H1 model, loop 162-165 closes the top of the ligand binding site. There are also errors in the modeled side chain conformations of the TM helices that intrude into the binding site. We have added more details in lines 293-295: "The poor performance was due to the inaccurate modeling of a loop containing residues 162-165 over the top of the ligand binding site as well as multiple errors in the conformation of side chains in the transmembrane helixes."